# The Mediating Roles of Lung Function Traits and Inflammatory Factors on the Associations between Measures of Obesity and Risk of Lower Respiratory Tract Infections: A Mendelian Randomization Study

**DOI:** 10.3390/healthcare12181882

**Published:** 2024-09-20

**Authors:** Xiaofeng Ma, Pan-Pan Zhu, Qian Yang, Yangbo Sun, Chun-Quan Ou, Li Li

**Affiliations:** 1State Key Laboratory of Organ Failure Research, Department of Biostatistics, Guangdong Provincial Key Laboratory of Tropical Disease Research, School of Public Health, Southern Medical University, Guangzhou 510515, China; maxiaofeng2022@i.smu.edu.cn (X.M.); zhupanpan112211@i.smu.edu.cn (P.-P.Z.); ocq@smu.edu.cn (C.-Q.O.); 2MRC Integrative Epidemiology Unit, University of Bristol, Bristol BS1 3NY, UK; qian.yang@bristol.ac.uk; 3Population Health Sciences, Bristol Medical School, University of Bristol, Bristol BS1 3NY, UK; 4Department of Preventive Medicine, The University of Tennessee Health Science Center, Memphis, TN 38163, USA; ysun80@uthsc.edu

**Keywords:** obesity, lower respiratory tract infections, lung function, inflammatory factors, mediation

## Abstract

Background: Identifying mediators between obesity-related traits and lower respiratory tract infections (LRTIs) would inform preventive and therapeutic strategies to reduce the burden of LRITs. We aimed to recognize whether lung function and inflammatory factors mediate their associations. Methods: We conducted a two-step, two-sample Mendelian randomization (MR) analysis. Two-sample MR was performed on (1) obesity-related traits (i.e., body mass index [BMI], waist circumference [WC], and waist-to-hip ratio [WHR]) and LRTIs (i.e., acute bronchitis, acute bronchiolitis, bronchiectasis, influenza, and pneumonia), (2) obesity-related traits and potential mediators, and (3) potential mediators and LRTIs. Next, two-step MR was applied to infer whether the mediation effects exist. Results: We found that C-reactive protein (CRP), interleukin-6 (IL-6), and forced expiratory volume in the first second (FEV1) mediated 32.59% (95% CI: 17.90%, 47.27%), 7.96% (95% CI: 1.79%, 14.14%), and 4.04% (95% CI: 0.34%, 7.74%) of the effect of BMI on pneumonia, and they mediated 26.90% (95% CI: 13.98%, 39.83%), 10.23% (95% CI: 2.72%, 17.73%), and 4.67% (95% CI: 0.25%, 9.09%) of the effect of WC on pneumonia, respectively. Additionally, CRP, forced vital capacity (FVC), and FEV1 mediated 18.66% (95% CI: 8.70%, 28.62%), 8.72% (95% CI: 1.86%, 15.58%), and 8.41% (95% CI: 2.77%, 14.06%) of the effect of BMI on acute bronchitis, and they mediated 19.96% (95% CI: 7.44%, 32.48%), 12.19% (95% CI: 2.00%, 22.39%), and 12.61% (95% CI: 2.94%, 22.29%) of the effect of WC on acute bronchitis, respectively. Conclusions: Health interventions linked to reducing inflammation and maintaining normal lung function could help mitigate the risk of obesity-related LRTIs.

## 1. Introduction

Lower respiratory tract infections (LRTIs) are a significant public health threat [1]. In 2019, there were 488.9 million cases of LRTIs, and LRTIs resulted in 2.4 million deaths globally [2], being the fourth cause of death [3]. The risk of LRTIs is affected by environmental factors as well as individual characteristics, such as age and health status (e.g., obesity) [4,5,6]. Over the past five years, the global incidence of obesity has continued to rise, becoming a public health issue of widespread concern [7]. According to the World Obesity Atlas, the number of obese individuals (body mass index [BMI] ≥ 30 kg/m^2^) aged over five years reached 988 million in 2020, accounting for 14% of the global population. It is estimated that, by 2035, this number will rise to 1.2 billion, representing 17% of the global population [8].

Previous studies have suggested associations between specific obesity-related traits and LRTIs. A systematic review indicated that obesity (BMI ≥ 30 kg/m^2^) was associated with an increased risk and severity of influenza and COVID-19 [9]. A prospective cohort study in the United States suggested that excessive weight gain was associated with an increased risk of community-acquired pneumonia [10]. A population-based prospective cohort study showed that both overweight (BMI > the 85th percentile) and obesity (BMI > the 95th percentile) were associated with an increased risk of bronchitis in adolescents [11]. Previous Mendelian randomization (MR) studies have revealed the effect of obesity (measured by BMI and waist circumference [WC]) on the risk of respiratory diseases [12,13,14]. Waist-to-hip ratio (WHR) is an important indicator of obesity, reflecting fat distribution [15]; however, prior studies merely reported the association of WHR with COVID-19 [16,17]. The associations with other types of LRTIs warrant further investigations. Clarifying the mediating pathways between obesity and LRTIs would inform preventive and therapeutic strategies to reduce the burden of LRTIs. Some lung function indicators and inflammatory factors have been linked to specific obesity-related traits and certain LRTIs, such as COVID-19 [18,19]. It is hypothesized that some lung function traits and inflammatory indicators play mediating roles between obesity and LRTIs. Further efforts are needed to identity the mediators of the associations between different obesity-related traits and various types of LRTIs and to what extent the mediators influence the associations.

The MR design utilizes genetic variations, typically single-nucleotide polymorphisms (SNPs), as instrumental variables to deduce the causal effect of exposure on the outcome. It effectively mitigates confounding bias and prevents reverse causality, provided that the core assumptions are satisfied [20]. Previous studies have applied the MR approach to examine the influential factors of infectious disease [21,22]. Herein, we conducted a two-step, two-sample MR to determine the roles of lung function indicators and inflammatory factors in the association between obesity and LRTIs. In this study, BMI, WC, and WHR were employed as the obesity-related traits [23]. We considered five common types of LRTIs, acute bronchitis, acute bronchiolitis, bronchiectasis, influenza, and pneumonia, as outcomes. Additionally, two lung function indicators and 13 inflammatory factors were considered as potential mediators.

## 2. Methods

### 2.1. Data Source

Summary statistics of the genome-wide association studies (GWAS) on BMI and WHR were obtained from a GWAS meta-analysis of 694,649 individuals of European ancestry [24]. Genetic associations with WC were extracted from a GWAS study of 462,166 UK Biobank participants of European ancestry. To avoid sample overlap for GWAS between genetically predicted obesity-related traits and genetically predicted potential mediators, we included an additional set of GWAS summary statistics for BMI, WC, and WHR from Genetic Investigation of Anthropometric Traits (GIANT), which did not include UK Biobank participants (Figure 1 and Table 1) [25,26].

Genetic associations with forced expiratory volume in the first second (FEV1) and forced vital capacity (FVC) were extracted from a GWAS study of 321,047 individuals of European ancestry [27]. For interleukin-1-receptor antagonist (IL-1Ra), interleukin-6 (IL-6), interleukin-8 (IL-8), interleukin-18 (IL-18), and interleukin-27 (IL-27), we used the data from a GWAS meta-analysis of 21,758 European ancestry participants [28]. Genetic associations with C-reactive protein (CRP) were obtained from a GWAS meta-analysis including more than 200,000 participants of European ancestry [29]. For glycoprotein acetyls (GlycA), we extracted the data from a GWAS study conducted among 115,082 UK Biobank participants of European ancestry [30]. For white blood cell count, basophil count, eosinophil count, neutrophil count, monocyte count, and lymphocyte count, we obtained the data from a GWAS meta-analysis of 563,085 European ancestry participants (Figure 1 and Table 1) [31].

We extracted genetic associations for LRTIs from the latest publicly available GWAS summary statistics database provided by FinnGen (Figure 1, Table 1, and Appendix A) [32]. All GWAS summary statistics used in this study were from European populations to ensure consistency in the genetic background. Ethical approval and participant informed consent were not required, since published summary statistics of the GWAS were used in this study. This study followed the Strengthening the Reporting of Genetic Association Studies (STREGA) reporting guideline.

### 2.2. Genetic Instrument Selection

We selected genetic instrumental variables for obesity-related traits, lung function indicators, and inflammatory factors based on the following criteria (Figure 1) [20]. First, we extracted SNPs robustly associated with the exposures (*p* < 5 × 10^−8^; if the number of SNPs was less than 10, then we further released the threshold to *p* < 5 × 10^−6^) [33]. Second, the clumping process was conducted to select independent SNPs so as to reduce bias due to linkage disequilibrium (*r*^2^ < 0.001, clumping distance = 10,000 kb). Third, we excluded SNPs with an *F* < 10 to avoid weak instrumental variable bias. The calculation of *F* is detailed in Appendix A. Fourth, we harmonized exposure and outcome data, removing palindromic SNPs with a minor allele frequency greater than 0.42 as recommended by the “TwoSampleMR” R package [34]. Finally, we employed Steiger filtering to exclude SNPs that do not explain more variance in the exposure than the outcome [35]. The selected SNPs are listed in Appendix A.

### 2.3. MR and Mediation Analysis

We conducted two-sample MR on (1) obesity-related traits and LRTIs, (2) obesity-related traits and potential mediators, and (3) potential mediators and LRTIs (Figure 1). We employed the inverse-variance weighted (IVW) in the main analysis and used weighted median, weighted mode, MR-Egger, and MR-Pleiotropy Residual Sum and Outlier (MR-PRESSO) in the sensitivity analyses (Appendix A). Cochran’s Q statistic was used to assess the heterogeneity of SNPs. If heterogeneity existed, the IVW random-effects model was employed. Conversely, a fixed-effects model was used. The MR-Egger intercept test was used to detect the presence of horizontal pleiotropy. The MR-PRESSO method was applied to identify possible outliers and correct the horizontal pleiotropy.

We applied the two-step MR to infer whether the considered lung function indicators and inflammatory factors mediate the associations between obesity-related traits and LRTIs (Figure 1). A factor was considered as a mediator of the association between an obesity-related trait and a type of LRTI if (1) the association between the obesity-related trait and the LRTI was statistically significant; (2) the association between the obesity-related trait and the potential mediator was statistically significant; (3) the association between the potential mediator and the LRTI was statistically significant; and (4) the 95% confidence interval (CI) of the proportion mediated did not contain zero. The proportion mediated was estimated as ${(\hat{\beta }}_{1}\times {\hat{\beta }}_{2})/\hat{\beta }\times 100\%$. ${\hat{\beta }}_{1}$ is the estimated change in a potential mediator (in standard deviation unit, except for CRP, whose unit was one-unit natural-log-transformed CRP) associated with per standard deviation (SD) increase in an obesity-related trait; ${\hat{\beta }}_{2}$ represents the estimated change in the log-odds of a type of LRTI associated with per SD increase in a potential mediator (except for CRP, whose unit was one-unit natural-log-transformed CRP); $\hat{\beta }$ means the estimated change in the log-odds of a type of LRTI associated with per SD increase in an obesity-related trait. The 95% CI of the mediation proportion was estimated using the delta method [36].

All analyses were conducted with R software version 4.3.1 (R Foundation for Statistical Computing). Packages for MR analyses included “TwoSampleMR” and “MRPRESSO” [37,38]. Two-sided *p* < 0.05 was considered statistically significant.

## 3. Results

With respect to the associations between obesity-related traits and LRTIs, elevated BMI and WC increased the risk of acute bronchitis and pneumonia. Likewise, enhanced BMI, WC, and WHR (odds ratio [OR] per SD = 1.14 [95% CI: 1.02, 1.27]) caused an increase in the risk of influenza. Nevertheless, BMI reduced the risk of bronchiectasis (Appendix A).

Concerning the associations of obesity-related traits with lung function indicators, FEV1 and FVC decreased with BMI, WC, and WHR. Regarding the associations with inflammatory factors, CRP and IL-6 increased with BMI, WC, and WHR. GlycA went up with WC. At the same time, GlycA, IL-1Ra, and IL-18 augmented with WHR. By contrast, monocyte count declined with BMI (Appendix A).

With regard to the associations of lung function and inflammatory factors with LRTIs, increased FEV1 (OR per SD = 0.80 [95% CI: 0.73, 0.88]) and FVC (OR per SD = 0.87 [95% CI: 0.79, 0.95]) lowered the risk of acute bronchitis, while elevated CRP augmented the risk. Enhanced FEV1 reduced the risk of acute bronchiolitis. Concerning pneumonia, enhanced FEV1 (OR per SD = 0.93 [95% CI: 0.88, 0.98]) and FVC (OR per SD = 0.94 [95% CI: 0.88, 1.00]) lowered the infection risk, whereas elevated CRP, IL-6, and monocyte count magnified the risk of pneumonia (Appendix A).

We identified three mediators (CRP, IL-6, and FEV1) of the associations of pneumonia with BMI and WC, given (1) the statistically significant effects of BMI and WC on pneumonia; (2) the associations of BMI and WC with the three factors; (3) the statistically significant effects of the three factors on pneumonia; and (4) the 95% CIs of mediation proportion that did not contain zero. It was estimated that CRP mediated 32.59% (95% CI: 17.90%, 47.27%) of the effect of BMI on pneumonia, which was higher than the proportion mediated by IL-6 (7.96% [95% CI: 1.79%, 14.14%]) and FEV1 (4.04% [95% CI: 0.34%, 7.74%]). Regarding the mediating pathways between WC and pneumonia, CRP, IL-6, and FEV1 mediated 26.90% (95% CI: 13.98%, 39.83%), 10.23% (95% CI: 2.72%, 17.73%), and 4.67% (95% CI: 0.25%, 9.09%) of the effect of WC on pneumonia, respectively (Figure 2 and Appendix A).

Based on the four criteria for the identification of a mediator, three factors (CRP, FVC, and FEV1) were identified as the mediators of the associations of BMI and WC on acute bronchitis. We found that 18.66% (95% CI: 8.70%, 28.62%), 8.72% (95% CI: 1.86%, 15.58%), and 8.41% (95% CI: 2.77%, 14.06%) of the effect of BMI on acute bronchitis were mediated by CRP, FVC, and FEV1, and these factors mediated 19.96% (95% CI: 7.44%, 32.48%), 12.19% (95% CI: 2.00%, 22.39%), and 12.61% (95% CI: 2.94%, 22.29%) of the effect of WC on acute bronchitis, respectively (Figure 3 and Appendix A).

## 4. Discussion

In this study, a two-step, two-sample MR approach was applied to identify the mediators of the association between obesity and LRTIs from two lung function indicators and 13 inflammatory factors. We identified four factors, FEV1, FVC, CRP, and IL-6, that mediated the effects of obesity on LRTIs. We found that elevated BMI and WC increased the risk of acute bronchitis, influenza, and pneumonia, and elevated WHR increased the risk of influenza. The findings were in accordance with a previous MR study, which reported the associations of respiratory diseases with BMI and WC [12]. Regarding the associations between obesity-related traits and bronchiectasis, it was observed that increased BMI reduced the risk of bronchiectasis, while the effect of WHR was statistically non-significant. Fat-free mass depletion was considered as a risk factor for increased morbidity and mortality in bronchiectasis [39]. Weight loss and loss of muscle mass are possibly associated with decreased respiratory muscle strength and can lead to poorer lung function among bronchiectasis patients. Increasing fat-free mass, not fat, may reduce the risk of bronchiectasis [40,41]. BMI does not distinguish between fat and fat-free mass [42], whereas WHR reflects the distribution of fat in the body, which might explain the difference between BMI–bronchiectasis and WHR–bronchiectasis associations.

Our findings suggested that obesity decreased FEV1 and FVC. Consistently, a cohort study conducted among adults aged 18–30 years in the United States revealed that FEV1 decreased with weight gain, and participants with higher weight experienced a more apparent decline in FVC [43]. Another study indicated that FEV1 and FVC decreased with obesity indices (i.e., WC, waist height ratio, WHR, and body fat) in an older Chinese cohort [44]. Additionally, a study showed that the lung function indicators (i.e., FEV1 and FVC) in obese individuals (BMI ≥ 30 kg/m^2^) were significantly lower than those in non-obese individuals [45]. We further discovered that (1) FEV1 and FVC mediated the association of acute bronchitis with BMI and WC; and (2) FEV1 mediated the association of pneumonia with BMI and WC. A possible explanation is that, in obesity, the deposition of fat in the thoracic and abdominal cavities leads to significant changes in the mechanical characteristics of the lungs and chest wall, potentially resulting in decreased lung compliance, airway narrowing, and increased respiratory system resistance, manifested as decreased FEV1 and FVC [46,47]. These physiological changes can disrupt normal ventilation and may lead to the retention of respiratory contents, such as mucus and bacteria [48], thereby increasing the risk of acute bronchitis and pneumonia.

It was found that CRP and IL-6 increased with obesity-related traits. Consistent with our study, a prospective cohort study suggested that, for each 1 kg increase in body weight, the average increase in CRP and IL-6 was 0.08 mg/L and 0.04 pg/mL, respectively [49]. A meta-analysis comprising 51 cross-sectional studies also revealed that obesity was associated with an increased level of CRP [50]. Prior studies demonstrated that serum levels of IL-6 and CRP were significantly higher in obese subjects (BMI ≥ 30 kg/m^2^) than non-obese subjects [51,52]. Regarding the association of CRP and IL-6 with the risk of LRTIs, a cohort study indicated that elevated IL-6 levels in patients with chronic obstructive pulmonary disease increased the risk of pneumonia [53]; another study showed that long-term pneumonia risk was associated with inflammatory serum markers (i.e., CRP) [54]. In addition, a study showed that elevated serum IL-6 and CRP levels were closely associated with the severity of pneumonia in COVID-19 patients [55]. Interestingly, we observed that CRP mediated the association of obesity (i.e., increased BMI and WC) with acute bronchitis and pneumonia, while IL-6 mediated the effects of BMI and WC on pneumonia. Compared to IL-6, CRP exhibited a higher mediation proportion. IL-6 is a pro-inflammatory cytokine synthesized by adipose tissue, endothelial cells, macrophages, and lymphocytes. It can activate immune cells, promoting the occurrence of inflammatory responses [56]. CRP is an acute-phase protein mainly synthesized in liver cells under the induction of inflammatory cytokines, especially IL-6 [57]. In the state of obesity, there is a significant increase in the number and size of adipocytes. These active adipocytes release substantial cytokines, such as IL-6 and tumor necrosis factor-alpha, which stimulate the increase in serum CRP level and promote inflammation [58,59]. Additionally, immune cell infiltration, particularly macrophages, occurs in the obese state, releasing inflammatory mediators that enhance the inflammatory response [60]. IL-6 stimulates mucus production by pulmonary epithelial cells [61], and the excess mucus physically obstructs the airways, increasing airflow resistance [62] and thereby potentially elevating the risk of LRTIs. Persistently high levels of IL-6 and CRP can result in an exaggerated or dysregulated immune response, disrupting lung homeostasis and further increasing the risk of pulmonary infections [63,64]. Influenza, an important type of LRTI, imposes a heavy disease burden on the whole population [65]. However, this study did not identify mediators between obesity and influenza. Further endeavors are required to elucidate the mediating mechanisms between obesity and influenza.

It is expected that targeting the disruption of pathways mediated by FVC, FEV1, CRP, and IL-6 can reduce the adverse impact of obesity on acute bronchitis and pneumonia, thereby alleviating the burden of LRTIs. Lung function tests aid in early detection and can inform management against lung function decline. Physical activity would help maintain or improve lung function [66]. Caloric restriction diet intervention is an option for the targeted population since it can effectively decrease CRP and IL-6 levels [67]. Furthermore, it is suggested to advocate diets rich in whole grains, dietary fiber, vegetables, fruits, fish, marine n-3 fatty acids, polyunsaturated fatty acids, vitamin C, vitamin E, and carotenoids for the targeted population, as previous studies have indicated that these diets can alleviate systemic inflammation [68].

Our study has several limitations. First, the GWAS data were derived from the European population, restricting the generalization of the findings to individuals of diverse ethnic backgrounds. Second, there was bias due to horizontal pleiotropy when assessing the association between WHR and acute bronchitis as well as the association between WHR and pneumonia, which prevented us from further exploring the potential mediation mechanisms. Third, the numbers of acute bronchiolitis and bronchiectasis cases were relatively small, possibly due to the lack of hospitalization records for the patients with mild symptoms, potentially resulting in insufficient statistical power. Fourth, we did not investigate the nonlinear association between obesity-related traits and LRTIs, since the validity of the nonlinear MR method is currently unclear [69]. Finally, we did not conduct subgroup analyses by age, sex, and obesity status; other potential mediators (e.g., presence of pathogenic bacteria or specific antibodies, neutrophil-to-lymphocyte ratio [70,71]) and the outcome of respiratory failure [72] were not examined in this study due to the lack of corresponding GWAS data. Further efforts with individual-level data are warranted to address these issues.

## 5. Conclusions

CRP, IL-6, and FEV1 play crucial intermediary roles in the association of obesity with pneumonia, and CRP, FVC, and FEV1 mediate the effect of obesity on acute bronchitis. Health interventions linked to reducing inflammation and maintaining normal lung function could help mitigate the risk of obesity-related LRTIs. Further efforts are warranted to identify other important mediators.

## Figures and Tables

**Figure 1 healthcare-12-01882-f001:**
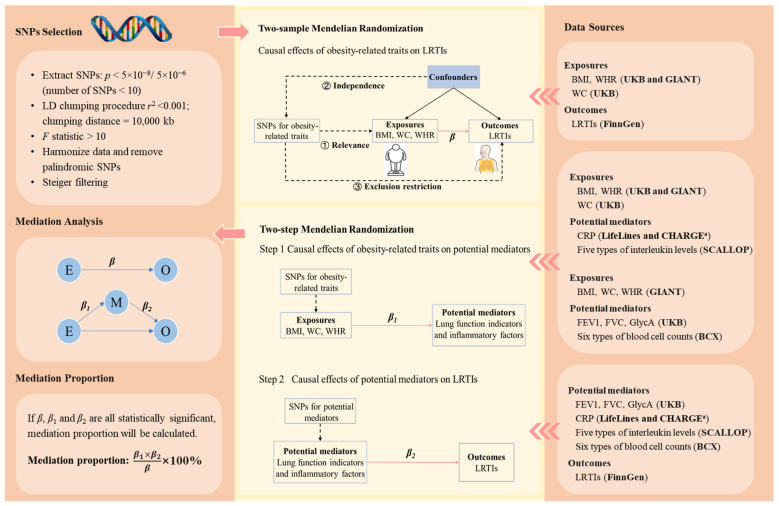
The schematic diagram of the study design. The three core assumptions of Mendelian randomization are (1) relevance assumption: the genetic variants are strongly associated with the exposure of interest; (2) independence assumption: there are no (unmeasured) confounders (e.g., population structure and assortative mating) between the genetic variants and outcomes of interest; and (3) exclusion restriction criteria: there is no pathway between the genetic variants and the outcome other than via the exposure of interest. *β* is the change in the log-odds of a type of LRTI associated with a standard deviation (SD) increase in an obesity-related trait. $\beta_{1}$ represents the change in a potential mediator (in standard deviation unit, except for CRP, whose unit was one-unit natural-log-transformed CRP) associated with per SD increase in an obesity-related trait. $\beta_{2}$ represents the change in the log-odds of a type of LRTI associated with per SD increase in a potential mediator (except for CRP, whose unit was one-unit natural-log-transformed CRP). The bold texts in the brackets in the right panel indicate the data sources. ^a^ Genetic associations with CRP were obtained from a meta-analysis of genome-wide association studies provided by LifeLines Cohort Study and CHARGE Inflammation Working Group consortia. The five types of interleukin levels were interleukin-1-receptor antagonist, interleukin-6, interleukin-8, interleukin-18, and interleukin-27 levels. The six blood cell counts were white blood cell count, basophil count, eosinophil count, neutrophil count, monocyte count, and lymphocyte count. Abbreviations: SNPs, single-nucleotide polymorphisms; LRTIs, lower respiratory tract infection; BMI, body mass index; WC, waist circumference; WHR, waist-to-hip ratio; FEV1, forced expiratory volume in the first second; FVC, forced vital capacity; CRP, C-reactive protein; GlycA, glycoprotein acetyls levels; UKB, UK Biobank; GIANT, Genetic Investigation of ANthropometric Traits; BCX, Blood Cell Consortium; SCALLOP, Systematic and Combined Analysis of Olink Proteins; CHARGE, Cohorts for Heart and Aging Research in Genomic Epidemiology.

**Figure 2 healthcare-12-01882-f002:**
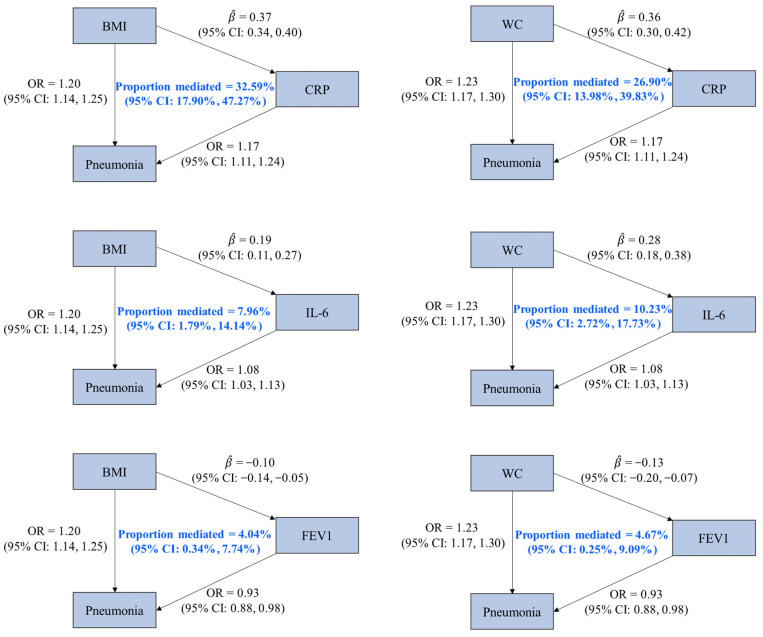
Associations among mediators, obesity-related traits, and pneumonia with mediation proportions. OR is the odds ratio of pneumonia associated with per standard deviation (SD) increase in an obesity-related trait or a potential mediator (except for CRP, whose unit was one-unit natural-log-transformed CRP). $\hat{\beta }$ represents the estimated change in a potential mediator (in one-unit natural-log-transformed CRP for CRP and in SD for others) associated with per SD increase in an obesity-related trait. Abbreviations: BMI, body mass index; WC, waist circumference; CRP, C-reactive protein; IL-6, interleukin-6; FEV1, forced expiratory volume in the first second; CI, confidence interval.

**Figure 3 healthcare-12-01882-f003:**
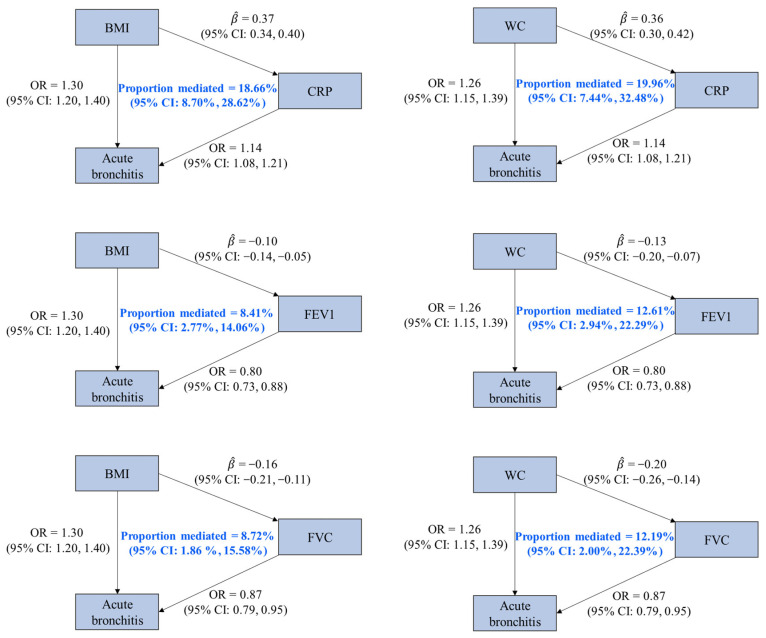
Associations among mediators, obesity-related traits, and acute bronchitis with mediation proportions. OR is the odds ratio of acute bronchitis associated with per standard deviation (SD) increase in an obesity-related trait or a potential mediator (except for CRP, whose unit was one-unit natural-log-transformed CRP). $\hat{\beta }$ represents the estimated change in a potential mediator (in one-unit natural-log-transformed CRP for CRP and in SD for others) associated with per SD increase in an obesity-related trait. Abbreviations: BMI, body mass index; WC, waist circumference; CRP, C-reactive protein; FEV1, forced expiratory volume in the first second; FVC, forced vital capacity; CI, confidence interval.

**Table 1 healthcare-12-01882-t001:** Details of data sources.

Traits	Consortium	ID	Unit	Sample Size
**Exposures**				
Body mass index	GIANT and UKB ^a^	None	SD	694,649
Body mass index	GIANT	ieu-a-2	SD	339,224
Waist circumference	UKB	ukb-b-9405	SD	462,166
Waist circumference	GIANT	ieu-a-61	SD	232,101
Waist-to-hip ratio	GIANT and UKB ^a^	None	SD	694,649
Waist-to-hip ratio	GIANT	ieu-a-72	SD	224,459
**Potential mediators**				
Forced expiratory volume in the first second	UKB	ebi-a-GCST007432	SD	321,047
Forced vital capacity	UKB	ebi-a-GCST007429	SD	321,047
C-reactive protein	LifeLines Cohort Study and CHARGE Inflammation Working Group	ieu-b-35	One-unit natural-log-transformed C-reactive protein	204,402
Interleukin-1-receptor antagonist	SCALLOP	ebi-a-GCST90012004	SD	21,758
Interleukin-6	SCALLOP	ebi-a-GCST90012005	SD	21,758
Interleukin-8	SCALLOP	ebi-a-GCST90011994	SD	21,758
Interleukin-18	SCALLOP	ebi-a-GCST90012024	SD	21,758
Interleukin-27	SCALLOP	ebi-a-GCST90012017	SD	21,758
Glycoprotein acetyls	UKB	ebi-a-GCST90092821	SD	115,082
White blood cell count	BCX	ieu-b-30	SD	563,085
Basophil count	BCX	ieu-b-29	SD	563,085
Monocyte count	BCX	ieu-b-31	SD	563,085
Lymphocyte count	BCX	ieu-b-32	SD	563,085
Eosinophil count	BCX	ieu-b-33	SD	563,085
Neutrophil count	BCX	ieu-b-34	SD	563,085
**Outcomes**				
Acute bronchitis	FinnGen	J10_BRONCHITIS	Not applicable	16,759 cases and 389,307 controls
Acute bronchiolitis	FinnGen	J10_BRONCHIOLITIS	Not applicable	2197 cases and 389,307 controls
Bronchiectasis	FinnGen	J10_BRONCHIECTASIS	Not applicable	2372 cases and 338,303 controls
Influenza	FinnGen	J10_INFLUENZA	Not applicable	9204 cases and 344,010 controls
Pneumonia	FinnGen	J10_PNEUMONIA	Not applicable	63,377 cases and 348,804 controls

^a^ The data originated from a meta-analysis combining the GWAS data provided by GIANT and UKB. Abbreviations: SD, standard deviation; GIANT, Genetic Investigation of ANthropometric Traits; UKB, UK Biobank; CHARGE, Cohorts for Heart and Aging Research in Genomic Epidemiology; SCALLOP, Systematic and Combined Analysis of Olink Proteins; BCX, Blood Cell Consortium.

## Data Availability

The datasets analyzed in the current study are available in the IEU OpenGwas platform (https://gwas.mrcieu.ac.uk/ [accessed on 1 January 2024]), the GIANT consortium (https://portals.broadinstitute.org/collaboration/giant/index.php/GIANT_consortium_data_files [accessed on 1 January 2024]), and the FinnGen consortium (https://www.finngen.fi/en [accessed on 1 January 2024]).

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
