# Peer review of "The Mediating Roles of Lung Function Traits and Inflammatory Factors on the Associations between Measures of Obesity and Risk of Lower Respiratory Tract Infections: A Mendelian Randomization Study"

_healthcare, 2024, doi:10.3390/healthcare12181882_

Round 1
Reviewer 1 Report
Comments and Suggestions for Authors
The manuscript titled "The mediating roles of lung function traits and inflammatory factors on the associations between measures of obesity and risk of lower respiratory tract infections: a Mendelian randomization study" presents a well-structured and scientifically robust approach to understanding the connections between obesity, lung function, inflammation, and lower respiratory tract infections (LRTIs). The study's aim to identify mediating factors such as lung function traits and inflammatory markers in the relationship between obesity and LRTIs is novel and clinically relevant. The study presents its results clearly, demonstrating the mediating effects of biological factors like C-reactive protein (CRP), interleukin-6 (IL-6), and forced expiratory volume (FEV1) on the relationship between obesity and LRTIs. These findings have practical implications for public health interventions. I have some suggestions for improvement:
1. With the use of large-scaled data collected from different sources (Table 1) and, presumably, different subjects of experiments, how would the authors make sure that the conclusions and relationships drawn are valid? This logic should be presented in the methodology section.
2. Why did the authors not include microbiological factors (e.g. presence of pathogenic bacteria or specific antibodies) as a potential mediator, since the presented LRTIs are related to microorganisms?
3. Would it be possible to compare the expression of significant potential mediators (CRP, IL-6, FEV1, etc) in obese and non-obese subjects? Is the expression of these proteins higher in obese people compared to non-obese ones naturally? The statement regarding this should be addressed in the discussion by considering existing related literature.
4. What are the potential limitations of this study?
Thank you.
Author Response
1. With the use of large-scaled data collected from different sources (Table 1) and, presumably, different subjects of experiments, how would the authors make sure that the conclusions and relationships drawn are valid? This logic should be presented in the methodology section.
Response: In this study, we conducted two-step, two-sample Mendelian Randomization (MR) analyses, where two-sample MR was performed separately for the associations between obesity-related traits and potential mediators, potential mediators and lower respiratory tract infections (LRTIs), and obesity-related traits and LRTIs. The data required for each two-sample MR analysis was derived from two populations with the same genetic background but without sample overlap, meaning that the subjects of the GWAS study in the two samples were not the same. We addressed the issue of sample overlap in the data source section of the Methods, ensuring that there was no sample overlap in each exposure-mediator-outcome pathway (lines 83-87 in the clean version). All GWAS summary statistics used in this study were from European populations to ensure consistency in the genetic background (lines 133-134). Moreover, in each MR analysis, we selected appropriate instrumental variables based on five criteria (lines 139-150). Finally, to ensure the robustness of results, we performed sensitivity analyses including weighted median, weighted mode, MR-Egger, and MR-Pleiotropy Residual Sum and Outlier (lines 154-156). The measures mentioned above are all taken to ensure that the conclusions drawn are valid.
2. Why did the authors not include microbiological factors (e.g., presence of pathogenic bacteria or specific antibodies) as a potential mediator, since the presented LRTIs are related to microorganisms?
Response: We thank the reviewer for this comment. We searched for data on microbiological factors related to lower respiratory tract infections (LRTIs) in public GWAS databases, including OpenGWAS and the GWAS Catalog. However, we did not find GWAS data for pathogens bacteria (e.g., pneumococcus, haemophilus influenzae, and staphylococcus aureus). In addition, the existing GWAS data for antibodies (e.g., IgA, IgG, and IgM levels) had insufficient sample size (approximately 1,000) to perform two-sample MR analyses. We also searched the UK Biobank but did not find relevant data. As a result, we are not able to include these factors in our analysis. We have acknowledged this as a limitation of our study in the revision (lines 321-323).
3. Would it be possible to compare the expression of significant potential mediators (CRP, IL-6, FEV1, etc) in obese and non-obese subjects? Is the expression of these proteins higher in obese people compared to non-obese ones naturally? The statement regarding this should be addressed in the discussion by considering existing related literature.
Response: Thank you for this suggestion. Prior studies showed that lung function indicators (i.e., FEV1 and FVC) were statistically significantly lower in obese individuals than in non-obese individuals [1] (lines 258-260), while serum levels of IL-6 and CRP in obese individuals were statistically significantly higher than those in non-obese individuals [2,3] (lines 273-275). However, our study used GWAS summary statistics to perform Mendelian randomization (MR). If we want to compare the expression of significant potential mediators in obese and non-obese subjects, we would need GWAS data for all exposures, potential mediators, and outcomes, stratified by obesity status. Unfortunately, the current public databases do not provide the data. We have mentioned this point in the limitations section (lines 320-321).
In addition, we conducted a preliminary analysis to examine the proportions of mediation effects in obese and non-obese subjects separately using the individual data from UK Biobank. We did not use MR in this analysis since the data of SNPs are not accessible for us currently. We estimated the associations between obesity-related traits (body mass index [BMI] and waist circumference [WC]) and potential mediators (i.e., forced expiratory volume in the first second [FEV1], forced vital capacity [FVC], and C-reactive protein [CRP]) (β1) using quantile regression models and assessed the associations between obesity-related traits and lower respiratory tract infections (LRTIs) (β), as well as potential mediators and LRTIs (β2), using Cox proportional hazards regression models. The LRTIs included acute bronchitis and pneumonia. Covariates included sociodemographic characteristics (i.e., age, sex, race, education level, employment status, and Townsend deprivation index), lifestyle factors (i.e., smoking status, drinking status, physical activity), environmental factors (i.e., particulate matter with aerodynamic diameters <2·5 μm [PM2.5]), and medical histories (i.e., hypertension, diabetes). Mediation analysis was considered only when the estimates of β, β1, and β2 were all statistically significant. We estimated the mediation effect using the product of coefficients method ( 1× 2), with the proportion mediated estimated as ( 1× 2)/ . The 95% confidence interval of the proportion mediated was estimated using the delta method. The results were shown in the Table below. The result showed that 95% confidence intervals of the proportions mediated for the obese group overlapped with the ones for the non-obese group. Further MR study will be performed with individual data of SNPs to provide more solid evidence on the mediating roles of lung function traits and inflammatory factors on the associations between measures of obesity and risk of LRTs in obese individuals as well as in non-obese individuals.
Table. The proportion mediated of each mediator between obesity-related traits and pneumonia, along with 95% confidence interval.
|
Exposure |
Potential mediator |
Outcome |
Obese |
Non-obese |
|
BMI |
Acute bronchitis |
FEV1 |
10.42% (0.23%, 21.00%) |
3.25% (0.92%, 6.00%) |
|
FVC |
/ |
5.25% (0.51%, 10.00%) |
||
|
Pneumonia |
FEV1 |
16.70% (11.06%, 22.00%) |
29.95% (11.21%, 49.00%) |
|
|
FVC |
15.73% (10.10%, 21.00%) |
47.52% (17.86%, 77.00%) |
||
|
CRP |
21.31% (14.40%, 28.00%) |
29.84% (11.35%, 48.00%) |
||
|
WC |
Acute bronchitis |
FEV1 |
/ |
1.71% (0.19%, 3.00%) |
|
Pneumonia |
FEV1 |
3.36% (2.02%, 5.00%) |
6.06% (2.93%, 9.00%) |
|
|
FVC |
2.11% (1.05%, 3.00%) |
/ |
||
|
CRP |
10.20% (7.63%, 13.00%) |
17.71% (10.91%, 25.00%) |
Note. BMI≥30 kg/m² was classified as obese, while <30 kg/m² was classified as non-obese. The quantile of the quantile regression model was specified as the median. The “/” indicated that the proportion mediated was not statistically significant. The number of participants were 234,809 for acute bronchitis (obese 52,298; non-obese 182,511), and 231,195 for pneumonia (obese 51,597; non-obese 179,598), respectively.
Abbreviations: FEV1, forced expiratory volume in the first second; FVC, forced vital capacity, CRP, C-reactive protein.
References
- Mala, K. Comparative study of pulmonary function tests in obese and non obese male individuals. Rajiv Gandhi University of Health Sciences (India) 2017, 30579732.
- Choi, J.; Joseph, L.; Pilote, L. Obesity and C-reactive protein in various populations: a systematic review and meta-analysis. Obes Rev 2013, 14, 232–244.
- Derosa, G.; Fogari, E.; D'Angelo, A.; Bianchi, L.; Bonaventura, A.; Romano, D.; Maffioli, P. Adipocytokine levels in obese and non-obese subjects: an observational study. Inflammation 2013, 36, 914–920.
4. What are the potential limitations of this study?
Response: The limitations of this study were elaborated in lines 311-325.
Reviewer 2 Report
Comments and Suggestions for Authors
Dear Authors,
I would like to thank you for your manuscript entitled " The mediating roles of lung function traits and inflammatory factors on the associations between measures of obesity and risk of lower respiratory tract infections: a Mendelian randomization study"
The manuscript is well written and the hypothesis is interesting.
I have one comment which is cruical:
CRP and IL-6 are mediators of inflammation and is commonly elevated in many diseases and cancers, which are not specifically related to the lung diseases. please find some scientific rational to link between obesity and IL-6/CRP, another link between lower respiratory tract infection and IL-6/CRP.
Thank you
Author Response
CRP and IL-6 are mediators of inflammation and is commonly elevated in many diseases and cancers, which are not specifically related to the lung diseases. please find some scientific rational to link between obesity and IL-6/CRP, another link between lower respiratory tract infection and IL-6/CRP.
Response: Thank you for your comments. So far, epidemiological studies have suggested the associations of obesity with IL-6 and CRP. A prospective cohort study suggested that for each 1-kilogram increase in body weight, the average increase in CRP and IL-6 was 0.08 mg/L and 0.04 pg/ml, respectively [1]. A meta-analysis comprising 51 cross-sectional studies also revealed that obesity was associated with increased level of CRP [2] (lines 270-273 in the clean version).
Regarding the association of CRP and IL-6 with the risk of lower respiratory tract infections (LRTIs), a cohort study indicated that elevated IL-6 levels in patients with chronic obstructive pulmonary disease increased the risk of pneumonia [3]; another study showed that long-term pneumonia risk was associated with inflammatory serum markers (i.e., CRP) [4]. In addition, a study showed that elevated serum IL-6 and CRP levels were closely associated with the severity of pneumonia in COVID-19 patients (lines 275-280) [5].
From a physiological mechanism perspective, in the state of obesity, there is a significant increase in the number and size of adipocytes. These active adipocytes release substantial cytokines, such as IL-6 and tumor necrosis factor-alpha, which stimulate the increase of serum CRP level and promote inflammation [6,7] (lines 287-290). IL-6 stimulates mucus production by pulmonary epithelial cells [8], and the excess mucus physically obstructs the airways, increasing airflow resistance [9] and thereby potentially elevating the risk of LRTIs. Persistently high levels of IL-6 and CRP can result in an exaggerated or dysregulated immune response, disrupting lung homeostasis and further increasing the risk of pulmonary infections [10,11] (lines 292-297).
References
- Fransson, E.I.; Batty, G.D.; Tabák, A.G.; Brunner, E.J.; Kumari, M.; Shipley, M.J.; Singh-Manoux, A.; Kivimäki, M. Association between change in body composition and change in inflammatory markers: an 11-year follow-up in the Whitehall II study. J Clin Endocrinol Metab 2010, 95, 5370–5374.
- Choi, J.; Joseph, L.; Pilote, L. Obesity and C-reactive protein in various populations: a systematic review and meta-analysis. Obes Rev 2013, 14, 232–244.
- Thomsen, M.; Dahl, M.; Lange, P.; Vestbo, J.; Nordestgaard, B.G. Inflammatory biomarkers and comorbidities in chronic obstructive pulmonary disease. Am J Respir Crit Care Med 2012, 186, 982–988.
- Lee, M.M.; Zuo, Y.; Steiling, K.; Mizgerd, J.P.; Kalesan, B.; Walkey, A.J. Clinical risk factors and blood protein biomarkers of 10-year pneumonia risk. PLoS One 2024, 19, e0296139.
- Sun, H.; Guo, P.; Zhang, L.; Wang, F. Serum Interleukin-6 Concentrations and the Severity of COVID-19 Pneumonia: A Retrospective Study at a Single Center in Bengbu City, Anhui Province, China, in January and February 2020. Med Sci Monit 2020, 26, e926941.
- Fonseca-Alaniz, M.H.; Takada, J.; Alonso-Vale, M.I.; Lima, F.B. Adipose tissue as an endocrine organ: from theory to practice. J Pediatr (Rio J) 2007, 83, Suppl 5, S192–S203.
- Visser, M.; Bouter, L.M.; McQuillan, G.M.; Wener, M.H.; Harris, T.B. Elevated C-reactive protein levels in overweight and obese adults. JAMA 1999, 282, 2131–2135.
- Neveu, W.A.; Allard, J.B.; Dienz, O.; Wargo, M.J.; Ciliberto, G.; Whittaker, L.A.; Rincon, M. IL-6 is required for airway mucus production induced by inhaled fungal allergens. J Immunol 2009, 183, 1732–1738.
- Agrawal, A.; Rengarajan, S.; Adler, K.B.; Ram, A.; Ghosh, B.; Fahim, M.; Dickey, B.F. Inhibition of mucin secretion with MARCKS-related peptide improves airway obstruction in a mouse model of asthma. J Appl Physiol 2007, 102, 399–405.
- Kony, S.; Zureik, M.; Driss, F.; Neukirch, C.; Leynaert, B.; Neukirch, F. Association of bronchial hyperresponsiveness and lung function with C-reactive protein (CRP): a population based study. Thorax. 2004, 59:892–896.
- Dawson, R.E.; Jenkins, B.J.; Saad, M.I. IL-6 family cytokines in respiratory health and disease. Cytokine 2021, 143:155520.

Reviewer 3 Report
Comments and Suggestions for Authors
This paper by Ma et al., enntitled “The mediating roles of lung function traits and inflammatory factors on the associations between measures of obesity and risk of lower respiratory tract infections: a Mendelian randomization study”, focused on the combined role of obesity and lung dysfunction in the pathogenetic chain of lower respiratory tract infections. Authors concluded that health interventions reducing inflammation and maintaining normal lung function could help mitigate the risk of obesity-related lower respiratory tract infections.Since inflammatory processes are usually associated with both obesity and lower respiratory tract infections, the strategy of assessing the combined mediating role of lung function and biomarkers of inflammation has no doubt important clinical implications.
Specific Comments
1. Authors show data regarding white blood cell (WBC) count and its cell percentages. It is well recognized in the Literature that neutrophil to lymphocyte ratio better than WBC and/or its cell percentages predict prognostic severity of Community Acquired Pneumonia (CAP) (de Jager et al., Plos One 2012), even in older patients (Cataudella et al., J Am Soc Geriatr 2017). Data on NLR should be reported and commented, particularly about the simulatneous occurrence of neutrophilia and lymphopenia. This is an important topic in presence of two pathogenetic factors of lower respiratory tract infections, such as obesity and lung dysfunction, where inflammation plays a crucial role.
2. Moreover, CRP not always encompasses all trajectories of prognosis in respiratory tract infections. For example, in Covid-19 patients, NLR was an independent predictor of mortality, resulted inversely related to PaO2/FiO2, while ICU admission was more significantly associated with CRP. Furthermore, by mediation analysis, it was shown that age, neutrophils, CRP, and lymphocytes are significantly and directly linked to PaO2/FiO2, while the influence of inflammation on PaO2/FiO2 , reflected by CRP, was also mediated by neutrophils (Regolo et al., J Clin Med 2023). Authors are therefore asked to focus on respiratory failure, a frequent complication of respiratory tract infections, particularly in obese patients.
3. Are there any data on biological and functional determinants of respiratory failure in obese patients with respiratory tract infections?
Author Response
1. Authors show data regarding white blood cell (WBC) count and its cell percentages. It is well recognized in the Literature that neutrophil to lymphocyte ratio better than WBC and/or its cell percentages predict prognostic severity of Community Acquired Pneumonia (CAP) (de Jager et al., Plos One 2012), even in older patients (Cataudella et al., J Am Soc Geriatr 2017). Data on NLR should be reported and commented, particularly about the simulatneous occurrence of neutrophilia and lymphopenia. This is an important topic in presence of two pathogenetic factors of lower respiratory tract infections, such as obesity and lung dysfunction, where inflammation plays a crucial role.
Response: We thank the reviewer for these insightful suggestions. We searched for data on neutrophil-to-lymphocyte ratio (NLR) and found that there is no publicly available GWAS summary statistics currently. Therefore, we conducted a preliminary analysis to assess the role of NLR using the individual data from UK Biobank. We did not use MR in this analysis, since the data of SNPs are not accessible for us currently. We estimated the associations between obesity-related traits (body mass index [BMI] and waist circumference [WC]) and NLR (β1) using quantile regression models and assessed the associations between obesity-related traits and lower respiratory tract infections (LRTIs) (β), as well as NLR and LRTIs (β2), using Cox proportional hazards regression models. The LRTIs included acute bronchitis, acute bronchiolitis, bronchiectasis, influenza, and pneumonia. Covariates included sociodemographic characteristics (i.e., age, sex, race, education level, employment status, and Townsend deprivation index), lifestyle factors (i.e., smoking status, drinking status, physical activity), environmental factors (i.e., particulate matter with aerodynamic diameters <2·5 μm [PM2.5]), and medical histories (i.e., hypertension, diabetes). Mediation analysis was considered only when the estimates of β, β1, and β2 were all statistically significant. We estimated the mediation effect using the product of coefficients method ( 1× 2), with the proportion mediated estimated as ( 1× 2)/ . The 95% confidence interval of the proportion mediated was estimated using the delta method. The results indicated that NLR played a mediating role between obesity-related traits and three outcomes (i.e., bronchiectasis, pneumonia, and respiratory failure) (Table below). Additional MR research will be conducted with individual data of SNPs to offer stronger evidence on the mediating roles of NLR on the associations between measures of obesity and risk of LRTIs.
Table. The proportion mediated by neutrophil-to-lymphocyte ratio between obesity-related traits and lower respiratory tract infection.
|
Exposure |
Outcome |
Mediation proportion (95% CI) |
|
BMI |
Bronchiectasis |
1.13% (0.67%, 1.59%) |
|
Pneumonia |
2.20% (1.39%, 3.01%) |
|
|
Respiratory failure |
1.14% (0.74%, 1.54%) |
|
|
WC |
Pneumonia |
0.67% (0.43%, 0.90%) |
|
Respiratory failure |
0.44% (0.28%, 0.60%) |
Note. The quantile of the quantile regression model was specified as the median. The number of participants were 235,791 for bronchiectasis, 231,195 for pneumonia (obese 51,597; non-obese 179,598), and 236,229 for respiratory failure, respectively.
2. Moreover, CRP not always encompasses all trajectories of prognosis in respiratory tract infections. For example, in Covid-19 patients, NLR was an independent predictor of mortality, resulted inversely related to PaO2/FiO2, while ICU admission was more significantly associated with CRP. Furthermore, by mediation analysis, it was shown that age, neutrophils, CRP, and lymphocytes are significantly and directly linked to PaO2/FiO2, while the influence of inflammation on PaO2/FiO2, reflected by CRP, was also mediated by neutrophils (Regolo et al., J Clin Med 2023). Authors are therefore asked to focus on respiratory failure, a frequent complication of respiratory tract infections, particularly in obese patients.
Response: Thank you for these suggestions. As mentioned in the response to the first comment, we analyzed the individual data from UK Biobank and the results for respiratory failure are showed in the Table below. The results indicated that lung function indicators and various inflammatory factors mediated the associations of three obesity-related traits with respiratory failure. Further MR study will be undertaken with individual data of SNPs to provide more solid evidence on the mediating roles of lung function traits and inflammatory factors on the associations between measures of obesity and risk of respiratory failure.
Table. The proportion mediated of each mediator between obesity-related traits and respiratory failure.
|
Exposure |
Potential mediator |
Mediation proportion (95% CI) |
|
BMI |
FEV1 |
36.32% (26.02%, 46.62%) |
|
FVC |
44.85% (31.81%, 57.89%) |
|
|
CRP |
19.81% (13.7%, 25.92%) |
|
|
White blood cell count |
22.27% (14.79%, 29.75%) |
|
|
Eosinophil count |
4.58% (1.46%, 7.69%) |
|
|
Monocyte count |
9.59% (6.10%, 13.09%) |
|
|
Neutrophil count |
17.05% (11.50%, 22.60%) |
|
|
NLR |
1.14% (0.74%, 1.54%) |
|
|
WC |
FEV1 |
12.51% (9.93%, 15.08%) |
|
FVC |
16.61% (13.14%, 20.09%) |
|
|
CRP |
11.01% (8.42%, 13.60%) |
|
|
White blood cell count |
13.6% (9.87%, 17.33%) |
|
|
Eosinophil count |
2.68% (0.74%, 4.63%) |
|
|
Monocyte count |
6.12% (4.20%, 8.04%) |
|
|
Neutrophil count |
10.92% (8.12%, 13.72%) |
|
|
NLR |
0.44% (0.28%, 0.60%) |
|
|
WHR |
FEV1 |
18.8% (15.49%, 22.11%) |
|
FVC |
19.6% (15.88%, 23.32%) |
|
|
CRP |
6.35% (4.98%, 7.71%) |
|
|
White blood cell count |
10.29% (7.58%, 13.01%) |
|
|
Eosinophil count |
1.84% (0.53%, 3.15%) |
|
|
Monocyte count |
4.56% (3.16%, 5.96%) |
|
|
Neutrophil count |
8.41% (6.35%, 10.47%) |
|
|
NLR |
0.23% (0.12%, 0.35%) |
Note. The quantile of the quantile regression model was specified as the median. The number of participants included in the analysis was 236,229. Abbreviations: BMI, body mass index; WC, waist circumference; WHR, waist-to-hip ratio; FEV1, forced expiratory volume in the first second; FVC, forced vital capacity, CRP, C-reactive protein; NLR, neutrophil-to-lymphocyte ratio.
3. Are there any data on biological and functional determinants of respiratory failure in obese patients with respiratory tract infections?
Response: We thank the reviewer for this comment. The mediators in the table above in the response to the second comment (i.e., lung function and inflammation markers) are biological and functional factors related to respiratory failure. We did not have other data on biological and functional determinants of respiratory failure.

Round 2
Reviewer 1 Report
Comments and Suggestions for Authors
The suggestions have been addressed in the revised manuscript
Author Response
The suggestions have been addressed in the revised manuscript
Response: We sincerely appreciate the reviewer’s suggestions.
Reviewer 3 Report
Comments and Suggestions for Authors
This paper was improved, but sill needs some additional changes in the reference list. In fact, references suggested in my report (see points 1 of your rebuttal (de Jager et al. 2012; Cataudella et al. 2017; see point 2 of your rebuttal: Regolo et al. 2023) are still missing in the reference list and must be added in it. So doing, this would support the strategy used to show additional analyses provided with the new version of the paper.
Author Response
This paper was improved, but still needs some additional changes in the reference list. In fact, references suggested in my report (see points 1 of your rebuttal (de Jager et al. 2012; Cataudella et al. 2017; see point 2 of your rebuttal: Regolo et al. 2023) are still missing in the reference list and must be added in it. So doing this would support the strategy used to show additional analyses provided with the new version of the paper.
Response: We thank the reviewer for this comment. We have added the three references mentioned by the reviewer in the reference list and cited them in the revision (line 322 in the clean version).